# Genome-Wide Analysis of DNA Demethylases in Land Plants and Their Expression Pattern in Rice

**DOI:** 10.3390/plants13152068

**Published:** 2024-07-26

**Authors:** Shengxin Mao, Jian Xiao, Yating Zhao, Jiaqi Hou, Lijia Li

**Affiliations:** State Key Laboratory of Hybrid Rice, College of Life Sciences, Wuhan University, Wuhan 430072, China; fuerst.mao@gmail.com (S.M.); 2021302041127@whu.edu.cn (J.X.);

**Keywords:** DNA demethylase, land plants, genome-wide analysis, plant development, stress response

## Abstract

DNA demethylation is a very important biochemical pathway regulating a group of biological processes, such as embryo development, fruit ripening, and response to stress. Despite the essential role of DNA demethylases, their evolutionary relationship and detailed biological functions in different land plants remain unclear. In this study, 48 DNA demethylases in 12 land plants were identified and classified. A phylogenetic tree was constructed to demonstrate the evolutionary relationships among these DNA demethylases, indicating how they are related across different species. Conserved domain, protein motif, and gene structure analysis showed that these 48 DNA demethylases fell into the presently identified four classes of DNA demethylases. Amino acid alignment revealed conserved catalytic sites and a previously less-studied protein region (referred to as domain A) within the DNA demethylases. An analysis showed a conserved pattern of gene duplication for DNA demethylases throughout their evolutionary history, suggesting that these genes had been maintained due to their importance. The examination of promoter cis-elements displayed potential signaling and regulating pathways of DNA demethylases. Furthermore, the expression profile was analyzed to investigate the physiological role of rice DNA demethylase in different developmental stages, in tissues, and in response to stress and various phytohormone signals. The findings offer a deeper insight into the functional regions of DNA demethylases and their evolutionary relationships, which can guide future research directions. Understanding the role of DNA demethylases can lead to improved plant stress resistance and contribute to the development of better crop and fruit varieties.

## 1. Introduction

DNA methylation, as a very important type of epigenetic modification, plays an essential role in many biological processes. Theoretically, epigenetic markers like cytosine DNA methylation lead to the alteration of chromatin structure [1]. In higher plants, according to studies on *Arabidopsis thaliana*, there are three main sites, CG, CHG, and asymmetric CHH (H referring to A, C, and T), for the existing methylated nucleotide 5-methyl-deoxycytidine (5-meC) in DNA molecules [2,3]. The main protein family responsible for DNA methylation establishment in plants is the DRM (Domains rearranged methyltransferase) family [4,5], whose mammalian orthologous counterpart is the DNMT3 (DNA methyltransferase 3) family [5]. For the maintenance of different methylated sequence contexts, plants utilize different enzymes for correspondent DNA contexts (for example, MET1 (Methyltransferase 1) for CG and CMT3 (Chromomethylase 3) for CHG) [6] and the methylation at the asymmetric site CHH (for example, CMT2 with DRM2) must be established de novo [5]. There are two potential DNA demethylation pathways: active and passive. DNA semiconservative replication allows for passive DNA demethylation to automatically occur in newly synthesized strands without the help of additional enzymes or factors. In newly synthesized DNA strands, all the bases lose any kind of modification like methylation. Thus, if there are no specific enzymes to create methylation on the newly synthesized strands and maintain DNA methylation level in the cell, the DNA methylation soon fades away after several rounds of DNA replication [7]. In active demethylation, it can take place independently from DNA replication but requires the participation of demethylases. Researchers have reported a couple of possible mechanisms of DNA demethylation [5,8]. However, all DNA demethylases including DME, ROS1, DML2, DML3 (Demeter for DME, Repressor of silencing, 1 for ROS1, and DME-like for DML) discovered in *A. thaliana* are reported to show a base excision-repair mechanism [9,10]. In the base excision-repair pathway, a specialized glycosylase cleaves the glycosidic bond, generating an AP (apurinic/apyrimidinic site) site; then, an endonuclease recognizes the AP site and removes the left deoxyribose. Eventually, DNA polymerase and ligase mend the generated gap [5].

The first reported enzyme for DNA demethylation in *A. thaliana* is ROS1 [9,11]. To be specific, the *A. thaliana ROS1* gene contains a highly conversed endonuclease III domain, which also features DNA glycosylase function [9]. *A. thaliana* DME bears a DNA glycoslyase/AP lyase bifunctional domain conserved to that of ROS1 [12]. According to current studies, those previously identified demethylases are all bifunctional enzymes [5,11]. Many DNA demethylases have been already identified in some other land plant species as well. For example, in *Oryza sativa*, the enzymes in the *DNG* (DNA glycosylase) gene family are reported to carry out DNA demethylase activity [13], as well as DML proteins in *A. thaliana*. In previous studies, orthologues of AtDML2 and AtROS1, which are OsDNG701 and OsDNG704, OsDNG702 (also known as OsROS1a) and OsDNG703, respectively, have already been identified [14,15]. However, there is no gene closely related to *AtDME* detected in rice [14]. An investigation into the physiological role of *DNG* family genes in plants demonstrated the indispensability of *DNG702* in rice gametophyte development [16] and other important functions of *DNG* family genes in gametes, zygotes [13] and pollen vegetative cells [17]. Many DNA demethylases like the *ZmROS1* gene in maize (*Zea mays*) have also been discovered and are believed to have important functions in maize endosperms [18]. These DNA demethylases play a crucial role in gene expression regulation and plant development. The function of ROS1 has been discovered to regulate trans-genes and some endogenous genes through the suppression of methylation-regulated silencing of genomes [19,20]. In *A. thaliana*, mutations in *ROS1* causing loss of activity may lead to many unwanted methylations of some genes and, thus, the unnecessary silencing of gene expression [9]. Expression of the *ROS1* gene is widespread within all plant tissues [9], and, in comparison, the *DME* gene is only transiently expressed in the central cell and synergid in the development of female gametophytes of dicots [10]. However, among four *A. thaliana* proteins, only the loss of function mutations in the *DME* gene generate severe growth defects in plants [10]. Mutations in the *DME* gene may bring about abnormal endosperm formation and deficient embryo development, consequently leading to seed abortion [21]. The DML2 and DML3 proteins can avoid DNA hypermethylation within vegetative tissues of *A. thaliana* [11]. These studies show that the process of DNA demethylation itself features several important biological meanings. First of all, demethylation is necessary for the early developmental stages of plants, without which impaired embryo development would occur [22,23]. Second, other reports have revealed that DNA demethylation has regulatory functions in the response of plants to abiotic stress, including drought [24], high concentration of salt [25], extreme temperature [26,27,28], and lack of nutrition [29,30]. The predicted mechanism of how DNA demethylases are related to stress response is that DNA demethylation at certain promoter sites triggers the transcription of downstream responding genes [31]. In addition, DNA demethylation also plays a role during the process of fruit maturation according to the proposed data from plants such as tomato and strawberry [32]. These reports demonstrate that investigations into DNA demethylation in plants have promising prospects in crop production improvement. However, recent studies on specific DNA demethylases mainly have mainly focused on *A. thaliana* and *O. sativa* [13,17,31]. 

Besides all the biological roles of DNA demethylase, it is also important to note that some DNA demethylases are reported to participate in plant–microbe interactions and the pathogen resistance of plants. To begin with, it was found that a protein called βC1, encoded by a satellite, affects the enzymatic activity of AtDME and NbROS1L in *Nicotiana benthamiana*, damaging the microbe-resistance of plants [33]. Another study discovered that *dme* mutants of *A. thaliana* have an increased susceptibility to both the bacterial infection of *Pseudomonas Syringae* (Pst DC3000) and the fungal infection of *Verticlium Dahliae*, and it concluded that DME plays a more prominent role in anti-pathogen response than other DNA demethylases, as well [34]. The role of ROS1 in plant resistance, especially in *A. thaliana*, has also been greatly investigated in many studies. For example, in contrast to the acquired resistance derived from plant–microbe interactions (or so-called systematic acquired resistance, SAR), a transcriptome analysis revealed that ROS1 controlled demethylation functions in transgenerational acquired resistance (TAR) against Pst DC3000 [35]. From the same research, the mechanism of the ROS1-dependent resistance was also proposed: a switch between demethylation and methylation on sialic acid (SA)-dependent gene alters their expression level [35]. Thus, it is very likely that the resistance is established based on the phytohormone, SA, and regulated pathways, similar to the case in abiotic resistance [36]. These reports, combined, indicate that in plants, DNA demethylases are largely involved in plant–pathogen interactions and are very likely to exert influences through phytohormone pathways. But whether the expression level of DNA demethylases is altered in response to a specific phytohormone remains unclear. 

At present, the development of bioinformatics approaches and the accumulation of bioinformation provides us with lots of methods and a tremendous resource pool to further explore more demethylase genes in a larger group of land plants. In this study, we identified and classified 48 genes in 12 land plants and then performed evolutionary analysis, amino acid alignment, gene structure, conserved domain, motif, promotor, and expression pattern analysis to discover the structure (catalytic sites and conserved domain), function (physiological role), and evolutionary pattern of DNA demethylase genes in different species. These data provide us with a better insight into further studies on DNA demethylases in land plants.

## 2. Materials and Methods

### 2.1. Identification of DNA Demethylases in 12 Land Plants

The genomes and annotated data of 12 land plants including 6 monocots [*Brachypodium distachyon* (v3.2), *Musa acuminata* (v1), *Oryza sativa* (v7.0), *Sorghum bicolor* (v3.1.1), *Setaria italica* (v2.2), and *Zea mays* (Refgen_v4)] and 6 eudicots [*Anacardium occidentale* (v0.9), *Arabidopsis thaliana* (Araport11), *Fragaria vesca* (v4.0.a2), *Prunus persica* (v2.1), *Solanum lycopersicum* (ITAG4.0), and *Vitis vinifera* (v2.1)] were downloaded from the Phytozome website (https://phytozome-next.jgi.doe.gov; accessed on 1 April 2024). The protein sequences of *A. thaliana* were used as queries to identify the sequences of potential DNA demethylases in these plants through the BlastP method. The longest transcripts of the BLAST results were preserved, while others were discarded. For the verification of the acquired protein sequences, the Batch Web CD-Search Tool on the NCBI CDD website (https://www.ncbi.nlm.nih.gov; accessed on 4 April 2024; [37]) was used to compare the conserved domains in BLAST results with control sequences in *A. thaliana*. Protein sequences with incomplete conserved domains were eliminated.

### 2.2. Sequence Alignment and Construction of the Phylogenetic Tree

All protein sequences of different DNA demethylase candidates in 12 land plants were aligned within a corresponding protein family using the MEGA11 software (Version 11.0.13). After the alignment of all sequences, the MEGA11 software was then used to construct phylogenetic trees of those protein sequences through neighbor-joining (NJ) methods based on the Poisson correction model, with pairwise deletion option. Meanwhile, the option of the bootstrap test was set to 1000 replicates [38]. A visualization of completed trees in NWK format was carried out by the iTOL website (https://itol.embl.de; accessed on 13 May 2024).

### 2.3. Protein Structure and Property Analysis

The conserved domains in our DNA demethylase candidates were analyzed on the NCBI conserved domain database (CDD) website using the Batch Web CD-Search Tool (https://www.ncbi.nlm.nih.gov; accessed on 4 April 2024; [37]). The MEME tool (http://meme-suite.org/tools/meme; accessed on 5 April 2024) on the MEME suite website was utilized to discover conserved motif patterns in those protein sequences. The results were visualized by the TBtools software (v2.084; [39]). The properties of target proteins, including pI and molecular weight, were predicted by the TBtools Seqcat function. The visualization of predicted peptide 3-dimensional structures was performed on the SWISS-MODEL website (https://swissmodel.expasy.org/interactive; accessed on 8 April 2024). The statistical data were graphed by the ChiPlot website (https://www.chiplot.online; accessed on 10 April 2024). 

### 2.4. Gene Structure Analysis

The structure of DNA methylase candidate genes was analyzed and visualized on the GSDS (Gene Structure Display Server; http://gsds.gao-lab.org; accessed on 4 April 2024) website. 

### 2.5. Gene Duplication Analysis of DNA Demethylase Families

The gene duplication analysis of candidate sequences was conducted on the MCScanX software (https://github.com/wyp1125/MCScanX, accessed on 4 April 2024) using the Blast P method, with the E-value option set to 1 × 10^−5^ [40]. The DnaSP6.0 software (v6.12.03) was applied to calculate both the nonsynonymous (K_a_) and synonymous (K_s_) substitutions and ratios of K_a_/K_s_ [41]. The divergence time calculation of duplicated DNA demethylase candidate gene pairs was based on the formula T = K_s_/(2 × 9.1 × 10^−9^) × 10^−6^ millions of years ago (MYA; [42]).

### 2.6. Promoter Analysis

To investigate into the cis-elements within promoter regions of candidate genes, the 2000 bp upstream of the gene translation start site sequences of corresponding genes was extracted from the plant genome sequences based on gff3 files. The identification of cis-elements was carried on the PlantCARE website [43]. Finally, the promoter patterns of targeted genes were visualized by the TBtools software (v2.084; [39]), and the statistical data were graphed by the ChiPlot website (https://www.chiplot.online; accessed on 11 April 2024).

### 2.7. Expression Pattern Analysis

Gene expression data of OsDNG704, OsDNG703, OsDML3a, OsDNG701, and OsROS1 were obtained from the RiceXPro databank (https://ricexpro.dna.affrc.go.jp; accessed on 12 April 2024), including those of different developmental stages and those responding to phytohormones. All expression arrays were subjected to 75 percentile normalization, and the Heatmap was visualized by TBtools [39]. 

## 3. Results

### 3.1. Identification and Classification of 48 DNA Demethylases in 12 Plants

The accumulation of growing amounts of plant genomic data enabled us to discover a large range of DNA demethylase candidates among lots of plant species. In previous studies, the major four DNA demethylases (ROS1, DME, DML2, and DML3) in *A. thaliana* have already been identified, and, thus, sequences of those genes and their corresponding proteins were set as positive controls and blast queries in our study. The genomic information of other 11 land plants including *A. occidentale*, *B. distachyon*, *F. vesca*, *M. acuminata*, *O. sativa*, *P. prunus*, *S. bicolor*, *S. lycopersicum*, *S. italica*, *V. vinifera,* and *Z. mays* was used as data sources for the identification of potential plant DNA demethylases. 

A previous study showed that enzymes performing DNA demethylase function contain three functional domains [44]. At present, the pattern and characteristics of DME domain and HhH-GPD domain (to be specific, Nth domain) are already stored on the NCBI CDD website, while information about N-terminal domain A remains unclear. Consequently, protein sequences, bearing both DME and glycosylase domains, were identified as DNA demethylases. Overall, we identified 48 genes (Table 1) which are predicted to be capable of demethylase activity. Among them, four *O. sativa* DNA demethylases (*OsDNG701*, *OsDNG702*, *OsDNG703,* and *OsDNG704*), three tomato (*S. lycopercicum*) genes (*SlDML*, *SlROS1L,* and *SlDML2*) [13], and four maize (*Z. mays*) genes (*ZmDNG105*, *ZmDNG102*, *ZmROS1,* and *ZmMDR1*) were already discovered. Then, all 48 identified protein sequences were subjected to MEGA software (Version 11.0.13) for alignment and phylogenetic analysis. Due to the uncertainty of the algorithm and model and far relationships between some genes, the tree of the genes has three primary clades instead of two clades as usual (Figure 1), and the bootstrap value above 50% is shown. The second clade is divided into two classes, which are designated *DMLa*, containing *AtDML2* and *AtROS1*, and *DME,* containing *AtDME*. The other two clades are classified into two groups, with the name of *DMLb* and *DMLc*. The phylogenetic tree demonstrates that the *DMLc* is the largest group among all four classes. One of the two clades in class *DMLb* is only composed of monocotyledonous genes, while the other comprises solely dicotyledonous genes (Figure 1). In addition, DNA demethylases in *M. acuminata* are only found in the *DMLc* class, while those of *V. vinifera* only exist in the *DMLa* and *DME* classes (Figure 1).

### 3.2. Recent WGD Events Generate Gene Diversification in Few Gene Pairs from Target DNA Demethylases

Gene paralogues are mainly derived from gene duplication events [45], which indicates the great contribution from gene duplication to gene and protein diversification. There are different types of evolutionary events that contribute to gene diversification, but in our study, we mainly focused on whole-genome duplication (WGD) events. MCScanX software (https://github.com/wyp1125/MCScanX, accessed on 4 April 2024) was used to discover gene duplicative pairs of DNA demethylases in 12 land plants’ genomes, and eventually, only 6 gene duplicative pairs were found within 48 DNA demethylase genes. Among them, only one pair was from monocots (*MaDML2* versus *MaDML3*) without a single one in Gramineae plants, while five were found in eudicots. This result indicates that, in monocots, or to be specific, in Gramineae, the diversification of DNA demethylases might mainly come from species differentiation instead of intraspecies duplication events. 

The estimated dates of duplication events in the six pairs of genes are relatively far away from the modern period (Table 2). Five pairs were estimated to be duplicated before 70 million years ago, when the differentiation of monocot and eudicot took place. One duplicative pair (*SlROS1L* vs. *SlDML*), however, emerged much later: about 34 million years ago.

### 3.3. Protein Structure and Motif Analysis Verify the Classification of DNA Demethylases 

To study the structural properties of DNA demethylase gene encoded proteins, all identified protein sequences were submitted to the Batch Web CD-Search Tool on NCBI. The result (Figure 2A) demonstrates that all demethylases possess the Nth domain (COG0177), and almost all have the RRM_DME Domain (pfam15628). The Nth domain is an endonuclease III domain, which is generally predicted to be responsible for DNA replication, recombination, and repair; while the RRM-DME domain is believed to be specified for DML proteins and is predicted to facilitate the recognition of specific DNA sequences through the guide of ssDNA or RNA [37,46]. Some of them also feature a Perm-CXXC domain (pfam15629) tandem with the C-terminal Nth domain. Incidentally, a very long and complete permease domain (Xan_ur_permease, cl23746) exists in a protein in *M. acuminata*, which may function as a transporter. This indicates that this protein might have multiple functions other than those of the DNA demethylase. The evolutionary relationships of DNA demethylases among species are seen from the phylogenetic analysis, as well. Proteins in DMLa containing AtROS1 and AtDML2 and DME including AtDME all feature a conserved pattern with three predicted domains (RRM_DME, Perm-CXXC, and Nth in order of from C-terminal to N-terminal). In contrast, the case in DMLb and DMLc is more complicated. Two enzymes (AtDML3 and AoDML) from eudicot in the DMLb group are expected to have the Perm-CXXC domain, while others in that class are not. One clade of DMLc (DMLcI) is only composed of two-domain proteins, and another mainly consists of proteins with three domains, which contain an additional Perm-CXXC domain apart from DME and Nth domains (Figure 2A). In DMLcII, all possess three domains, except OsDNG703 and ZmDNGI02. These results not only further verify our construction of the phylogenetic tree, but also indicate some subtle details of potential evolutionary events, such as gene duplication and functional differentiation. 

The information of the previously reported domain A, required for DNA binding and 5 meC excision [44], has not been thoroughly studied and is, thus, presently not stored in the database on NCBI. The motif analysis, however, shows a possible domain A region. The MEME suite was used to discover the most likely 20 motifs in our identified protein sequences, and all 20 potential motifs were visualized by TBtools. Most identified DNA demethylase proteins share a very similar motif composition, with motif pattern 16-7-3, at the N-terminal side of the Nth domain region (Figure 2B), although some proteins in the DMLb class do not bear motif 16, solely featuring tandem motif 7 and motif 3. One protein in *Z. mays* is the only exception with neither 16-7-3 nor 7-3 patterns (Figure 2B). According to previous research [44], this region containing such motif pattern is very likely to be that so called domain A due to its position in the protein sequence. From the diagram, we can also conclude that the Nth domain of DNA demethylases is highly conserved in evolution because practically all Nth domains in different proteins have a motif composition in the order of 20-9-5-1-2-8. At the C-terminal end of most DNA demethylases in classes DMLa, DMLc, and DME, there exists a longer motif pattern with nine tandem motifs (19-14-13-11-12-17-4-10-6) (Figure 2B). This region covers both the DME domain and the small Perm-CXXC domain, and also even includes a large part of IDR (intrinsic disordered region) between Nth and DME. In fact, there is a small gap between motifs 12 and 17 in most DMLa, DME, and DMLc proteins, which separates this region featuring long conserved motif patterns into two parts. And from the sequence and domain information, the latter part (motifs 17-4-10-6) is the predicted Perm-CXXC and DME regions. In DMLb, most proteins, in comparison, lack an entire 9-motif composition pattern, but all show a 4-10-6 motif pattern, which very likely comprises the DME domain at the C-terminal end of our identified DNA demethylases (Figure 2B). Moreover, motif 19 is possessed by all DNA demethylases in IDR, which may play an unknown role in the enzyme activity or regulation of the protein itself.

### 3.4. Gene Structure Analysis Shows the Exon–Intron Patterns of DNA Demethylases

An investigation of the gene structure shows some interesting trends in our identified DNA demethylase genes. The most common pattern in one clade of *DMLc* class (*DMLcII*) is two long exons at the 5′ end with a small exon between them, followed by six tiny exons, one intermediate exon, and then six to seven small exons again (Figure 3). In comparison, in another clade of this class (*DMLcI*), the two long exons are separately divided into a short exon and a long exon, while the rest have the same pattern. However, *M. acuminata* genes in *DMLc* bear an exon–intron pattern with some differences. In *DMLcII*, the second long exon in two *M. acuminata* genes is totally cut up by introns into a group of 7 small exons, and for that in subclass 1, it has additional 11 small exons at the 3′ end apart from the cut second large exons. The result of extra exons in the *M. acuminata DMLcI* gene is consistent with our previous domain and motif analysis (Figure 2A,B), meaning that these extra exons encode a different functional domain (permease domain). The pattern of small exons at the 3′ end of the *DMLa* and *DME* genes is quite similar to the pattern of the *DMLc* genes. But most of their first large exons are flanked by two additional small exons. The pattern of *DMLb* is like a combination of both previously mentioned patterns. One clade has almost exactly the same pattern as *DMLcII*, with shorter exons, while another has extra flanking exons as *DME* and *DMLa*. In conclusion, this similar exon–intron pattern may indicate the very conserved evolutionary relationship of DNA demethylase genes. 

### 3.5. Sequences Alignment Demonstrates the Conserved Enzymatic Activity Sites

The potential enzymatic activity sites of glycosylases have already been identified in AtDME and AtROS1 domains [12,44]. Three sites, a lysine residue (Lys-1544) and an aspartic residue (Asp-1562) in AtDME (a transcript version with a protein length of 1987 aa), and four cysteines constituting a 4Fe-4S cluster nearby, are crucial to the glycosylase/lyase activity of the glycosylase domain [12]. These sites are predicted to be highly conserved in glycosylase domains of other enzymes in humans [12] and bacteria [12,44]. In addition, the investigation into detailed mechanisms of ROS1 shows that some hydrophobic residues form a pocket to accommodate 5 meC [47]. In our analysis, it is shown that all of the identified DNA demethylases have the same conserved lysine and aspartate residues in the glycosylase domain (Figure 4) and almost all have the 4Fe-4S cluster, only with one exception of AoDML which features a 3Fe-3S cluster. Thus, the result indicates that those sites are essential for glycosylase domain function, which is also supported by empirical data [12,47]. However, according to our analysis, there are many other completely conserved sites that are sequentially proximate to those three sites in aligned protein sequences. These sites might constitute the enzyme pocket which functions to accommodate the substrate 5-meC of DNA molecules and recognizes the DNA molecule.

Domain A (recognized as protein region from 980 to 1080 of a AtDME transcript version with a protein length of 1987aa) in AtDME is composed of a mixed charge cluster, which is likely to function as DNA binding sites and facilitates 5 meC excision, according to the previous reports [44,47], and the result of the sequence alignment of 48 DNA demethylases is in agreement with that in AtDME (Figure 5A). In this study, a statistical analysis demonstrates that among all Demeter-like proteins (only with the exception of ZmMDR1), the charge amino acid composition in a potential Domain A region is significantly higher than the whole protein or common proteins (Rice Rubisco) in plants (Figure 5A), and it is in correspondence with that of the DNA binding region of the rice histone-fold domain (HFD_H4). The predicted three-dimensional structure of this region shows that this region folds into a regular helix–turn–helix DNA binding structure (taking AtDME and SlDML2 as examples) (Figure 5C,D), which is very similar to that of HFD_H4 (Figure 5B). This conserved pattern may work as a supplement for the assumption that domain A may be necessary for protein–DNA interactions between DNA and DNA demethylases.

### 3.6. Cis-Element Analysis Shows Implication for the Future Research of DNA Demethylases in Plants 

According to previous studies, DNA demethylation mainly functions in plants’ early development [10,21] and response to abiotic and biotic stresses [25,27,29]; thus our study is focused on cis-elements that participate in those areas, like elements that are susceptible to phytohormone. Among all the targeted cis-elements that showed up in promoter regions of 48 genes, the most abundant class is that responsive to light. In total, we discovered 528 light-responsive elements with 27 different types in the promoter region of 48 DNA demethylases, which include G-box, Box 4, GT1-motif, etc. Elements responsive to phytohormone were also very common in the promoter region of our genes. We identified a total 11 kinds of different cis-elements that correspond to 5 major phytohormones (Methyl Jasmonate-MeJA, Gibberellic acid-GA, Abscisic acid-ABA, Salicylic acid-SA, and Auxin; Table 3). Among them, elements responsive to ABA (ABRE element with 136 times) and GA have the highest frequency (CGTCA-motif and TGACG-motif both with 74 times; Table 3 and Figure 6C). In addition, we also discovered some elements specifically responsible for stress and plant development. For example, ARE elements that respond to anaerobic induction were widespread among 48 genes with only a few exceptions. Elements induced by low temperature (LTR) and drought (MBS) were less common but still appeared in those promoter regions with a frequency of 44 times for MBS and 31 times for LTR, which cannot be ignored. Elements such as CAT-Box, GCN4_motif, and RY-element corresponding to meristem development, endosperm formation, and seed development were also discovered, which confirm the role of DNA demethylation in plant development (Table 3 and Figure 6A–C). 

### 3.7. Expression Patterns of Rice DNA Demethylase Genes in Different Developmental Stages and Tissues and in Response to Phytohormones

Characteristics of gene expression in different organs and tissues and through different developmental stages empower us to thoroughly understand gene function [48]. Based on data collected from rice plants grown in standard conditions and treated by specific factors, including phytohormones and pathogen inoculation, the construction of standard expression profiles enables us to find subtle clues for the functional differentiation of different genes [49]. This analysis demonstrates that, overall, *OsDML3a* and *OsDNG701* feature obviously higher expression levels than *OsDNG704*, *OsDNG703,* and *OsROS1*, indicating the potentially more significant role in the organism of the former two (Figure 7B,D). Through different rice tissues, the expression level of *OsDML3a* and *OsDNG701* in some reproductive tissues, such as anther and pistil, is generally higher than in vegetative organs (root, leaf blade, etc.) and embryo and endosperm, while the expression pattern of *OsDNG704*, *OsDNG703,* and *OsROS1* is reversed (Figure 7A). However, the expression alteration of *OsDML*s during different reproductive tissue developmental stages is not obvious (Figure 7A).

From the results, it is clearly illustrated that the expression level of all five rice DNA demethylases shows a relative accordant pattern in response to different phytohormones. In general, the expression of these DNA demethylases is upregulated both in shoots and roots under the treatment of JA, BR (brassinolide), and GA (with the only apparent exception of *OsDML3a* in response to JA in the shoot) (Figure 7C). In comparison, the expression of five rice DNA demethylase genes is downregulated or remains at a low level in response to the treatment of CK (cytokinin), IAA (indoleacetic acid), and ABA in shoots and roots (Figure 7C). However, exceptions still exist. In rice shoots, the expression level of *OsDNG701* and *OsROS1* is high rather than low in response to CK, whilst the expression of *OsDML3a*, *OsDNG704*, and *OsDNG703* is upregulated instead of downregulated in rice roots under the treatment of IAA (Figure 7C).

## 4. Discussion

### 4.1. Construction of Phylogenetic Tree and Further Verification Lead to New Classes of DNA Demethylases

In previous studies, the phylogenetic analysis of plant DNA demethylases was either restricted to a few species or only roughly classified as the potential DNA demethylase genes without further verification and analysis [31,50]. But, in this study, the 48 genes are identified and clearly classified into four distinct classes (*DME*, *DMLa/b/c*; Figure 1). Within the same class, DNA demethylases show obvious consensus, such as common protein properties like pI value (Table 4), conserved motif composition (Figure 2B), and gene structure (Figure 3). For example, the class DMLc has the longest exons (Figure 3), compared to DMLa, DMLb, and DME, at the 5′ end of the DNA molecule. In contrast, the DMLb class genes at that end have exons with the shortest length. This indicates that the exon and intron distribution pattern within a single class is relatively conserved. However, the expected gene structure generated from bioinformatical algorithm may need further modification through experiments. 

The construction of a phylogenetic tree also demonstrates an interesting result that two classes are eudicot-specific (*DMLa* and *DME*) (Table 1; Figure 1) and one is restricted in monocots (*DMLc*) (Table 1). The conclusion that genes closely related to DME are eudicot-specific is also supported by a published report [50]. From the phylogenetic tree (Figure 1), DNA demethylases from *F. vesca* and *P. persica* in classes *DMLa*, *DME*, and *DMLbI* are all closely related and occur in pairs (FvDME with PpDME2 and PpDME in class *DME*, PpDML2 with FvDML and FvDML4 in class *DMLa*, and PpDML with FvDML3 and FvDML2 in class *DMLbI*; Figure 1). These results are plausible because the two species come from one family (Table 1); thus, they have a close evolutionary relationship. This indicates that DNA demethylases from the two species were derived from evolutionary events slightly before the divergence between *F. vesca* and *P. persica* about 69 million years ago [51] or may just come from the species differentiation of *F. vesca* and *P. persica*. There are two main subclasses within *DMLc*, which both contain different DNA demethylases from all six monocots. The result may indicate that DNA demethylases of this class in the same species are paralogous and underwent duplication in their evolution history. However, just as the case of DNA demethylases in *F. vesca* and *P. persica*, some DNA demethylases from *S. bicolor* and *Z. mays* also occur in pairs in class DMLc (for example, ZmROS1 with SbDML5 in *DMLcI* and ZmDNG102 with SbDML2 in *DMLcII*). Among all the six monocots, *Z. mays* and *S. bicolor* do have the closest relationship according to the estimated divergence time using the Timetree website [51]. Thus, we may locate the evolutionary origin of these DNA demethylases at a date of about 16 million years ago (the divergence time of *Z. mays* and *S. bicolor*; [51]). From the observation that *DMLbII* is only composed of monocotyledonous genes, while *DMLbI* comprises solely dicotyledonous genes, we can conclude that these two clades may share orthologs and are derived from relative ancient species differentiation. The classification provides a new way for further discovery of more plant DNA demethylases through exploring the orthologs of already identified genes in the corresponding class. In addition, it also equips researchers with a deep insight into the evolution history of DNA demethylases or even species themselves.

### 4.2. Multiple Sequence Alignment and Domain and Motif Analysis Discovered a New Functional Region of DNA Demethylases

Although domain A of DNA demethylases proximate to N-terminal of the enzyme has been long considered as functional [44], it has not been profoundly understood. Some conserved sites sequentially near that region may participate in the formation of substrates accommodating pockets [47] of glycosylase and were, thus, investigated, but other poorly conserved amino acids sites were ignored in many studies. However, motif analysis shows that a conserved motif pattern and composition appear in the region of all DNA demethylases of 12 land plants, forming a potential domain. A high frequency of the appearance of charge amino acids in this region also indicates that this region is very likely to play a role in DNA binding (Figure 5A) [44], which may play an important role in the stabilization or recognition of DNA substrates. The visualization of the predicted protein 3-dimensional structure of the potential domain A region also indicates that this DNA binding region is very likely to have a traditional helix-turn-helix pattern (Figure 5B–D).

The high frequency of the appearance of red color-designated conserved amino acid sites in the glycosylase/lyase domain region in DNA demethylases (Figure 4) also demonstrates that the bifunctional domain is highly conserved during the species differentiational events in evolutionary history, and the function of a single amino acid is relatively irreplaceable. 

### 4.3. Cis-Element Analysis Shows the Future Investigation Field of the Role of DNA Demethylase in Plant Development Regulation and Stress Response 

It has already been known that the expression of DNA demethylases is regulated by DNA methylation levels [52,53,54]. Some studies have also revealed the function of DNA demethylases in plant development [10,55], fruit ripening [56], and response to extreme conditions [36]. However, these studies mainly focused on the pathway and reaction regulated by DNA demethylation, or the influence of methylation on the expression of the DNA demethylase. But, by looking into promoter regions of identified DNA demethylase genes, it is found that there are some of cis-elements related to many different conditions and phytohormones (Table 2), which indicates the latent and possible direct regulation by phytohormones and external factors. Consequently, this provides a research direction: for example, an investigation into the possible mechanism of condition-induced transcription activation or suppression of DNA demethylase genes. Due to the considerable frequency of the appearance of anaerobic and anoxic resistance-responsible cis-elements (ARE and GC-motif) in the promoter region (Figure 6C), the DNA demethylases may, to some extent, be predicted to play a plausible role in plant tolerance against waterlogging stress and the rescue of drowned roots [57]. Moreover, since phytohormones such as auxin and GA also involve the rescue of waterlogging conditions [58], DNA demethtylases may participate in such response indirectly. The frequent appearance of MeJA, which, in rice and other plant species, acts in defense of fungal pathogens [59,60], regulating cis-elements (Figure 6C), proves that the DNA demethylases are very likely to have the potential to be involved in anti-fungi processes. As mentioned previously, there is a recent report proving that *AtROS1* may participate in the anti-pathogen process of plants through the SA regulated pathways [35], and our results may provide a new research direction for the study of the role of another phytohormone, MeJA, in the DNA demethylase-regulated pathway against fungal pathogens (Figure 6B,C; Table 3). The abundance of ABA-responsible elements also indicates that DNA demethylases may participate in plant resistance to changing environments, in accordance with a previous study [31].

### 4.4. Expression Patterns Analysis of OsDMLs Shows the Separated Role of Different Rice DNA Demethylases in Response to Specific Phytohormones and in Different Tissues

The expression profile of rice DML genes of different developmental stages demonstrates that DNA demethylases in rice are very likely to undergo functional diversification (Figure 7A). *OsDML3a* and *OsDNG701* feature a higher expression level in reproductive tissues (anther, pistil, etc.) (Figure 7A), which indicates that this pair of genes may participate in the formation of gamete and flowering. In comparison, the expression of *OsDNG704* and *OsDNG703* is higher in the embryo and endosperm, leading to a conclusion that they may be involved in the regulation of the embryo and endosperm’s development. Thus, such DNA demethylase genes may perform a very similar function as *DME* in *A. thaliana,* which is mainly expressed in reproductive tissues [61]. Furthermore, *OsROS1* is expressed more in vegetative organs, and this consequently shows that it may mainly function in response to stress and changing environments. The fact that *OsROS1* has a considerable expression level in vegetative tissues is similar to the case of the *ROS1* gene in *A. thaliana*. Previous reports show that some DNA demethylases like *ROS1* in *A. thaliana* are expressed mainly in vegetative tissues and play an important role in disease and stress response [31,62]. These results show that *OsROS1* might have a very similar role as its *A. thaliana* counterparts in disease and stress resistance.

The next heatmap (Figure 7C) illustrates that the expression level of DNA demethylases in rice do respond to several specific phytohormone treatments and, thus, may give a hint that the results are in accordance with the results from cis-element analysis (Figure 6; Table 3). It is noticed that, especially for JA (or MeJA), the high expression level of rice DNA demethylases seen from the heatmap may be the consequence of the direct regulation by those cis-elements in their promoter regions (except *OsDNG703*, which does not contain related cis-elements) (Figure 7C; Table 3). Thus, these results may further verify the assumption that rice DNA demethylases are involved in fungal pathogen resistance. There are no recent detailed investigations into specific DNA demethylases in response to a particular phytohormone; however, there is research demonstrating that the level of DNA methylation and demethylation is altered in response to phytohormone [63]. Moreover, the data collected from predictive bioinformatic analysis still need further verification through wet experiments. For example, there is a considerable amount of cis-elements responding to phytohormone ABA in the promoter region of *OsROS1* (Table 3), but it is seen from the expression profile that the expression of *OsROS1* remains low under the treatment of ABA (Figure 7C). Such results may derive from more complicated situations or regulations, which must be further studied through detailed mechanism investigations.

## 5. Conclusions

In this study, 48 DNA demethylases in 12 land plants including *A. thaliana*, *A. occidentale*, *B. distachyon*, *F. vesca*, *M. acuminata*, *O. sativa*, *P. prunus*, *S. bicolor*, *S. lycopersicum*, *S. italica*, *V. vinifera,* and *Z. mays* were compared and analyzed through bioinformatical tools. Their evolutionary relationships were uncovered through the construction of a phylogenetic tree, and four main classes (DME and DMLa/b/c) were established. Among them, class DMLb is monocot-specific. Conserved domain, protein motif pattern, and gene structure analyses further verified the similarities and relationships of DNA demethylases within the same class. The sequence alignment of DNA demethylase proteins showed a conserved and highly structured region, which was believed to be domain A and might have a role in DNA binding, facilitating the recognition or stabilization of DNA–protein interaction. The cis-element analysis of the promoter region in DNA demethylase genes provides future directions in studying the role of DNA demethylases in stress and defense response or in response to phytohormone, while the expression profile construction in rice, to some extent, supports the general role of DNA demethylases in organisms through pointing out their expression alternation and difference in response to different phytohormones and in different tissues. And for better crop varieties creation, future investigations may be conducted into crops with an enhanced function of DNA demethylases.

## Figures and Tables

**Figure 1 plants-13-02068-f001:**
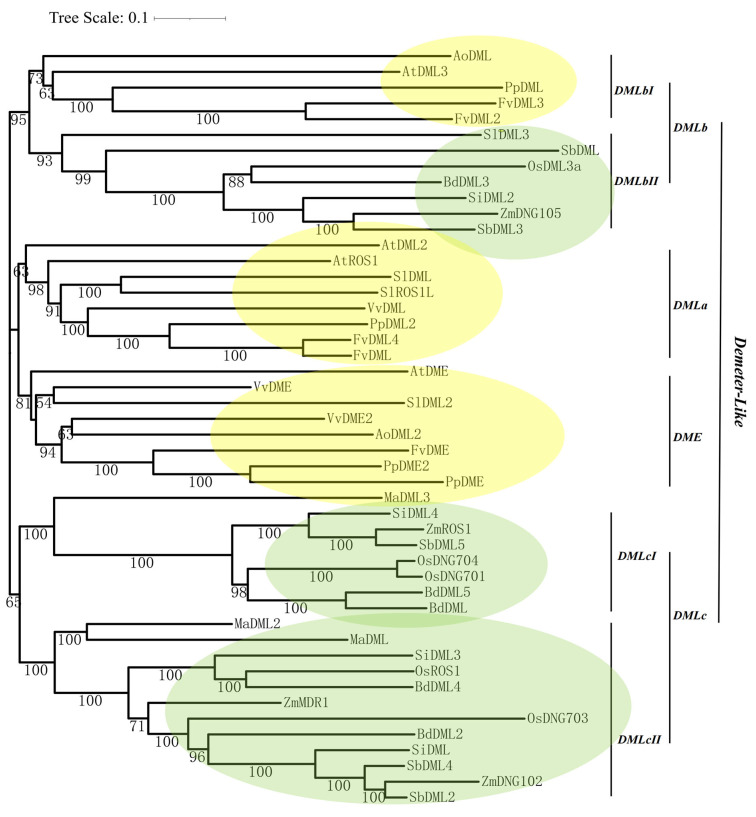
Phylogenetic analysis of Demeter-like proteins in 12 land plants (Unrooted NJ tree of 48 Demeter-like proteins). Bootstrap test values >50 are shown next to the branch. Classes and clades are shown on the right side. The green color indicates that the class is monocot-specific. The yellow color indicates that the class is eudicot-specific. The scale bar stands for the number of nucleotide replacements per site. The corresponding gene Locus ID and name are shown in Appendix A.

**Figure 2 plants-13-02068-f002:**
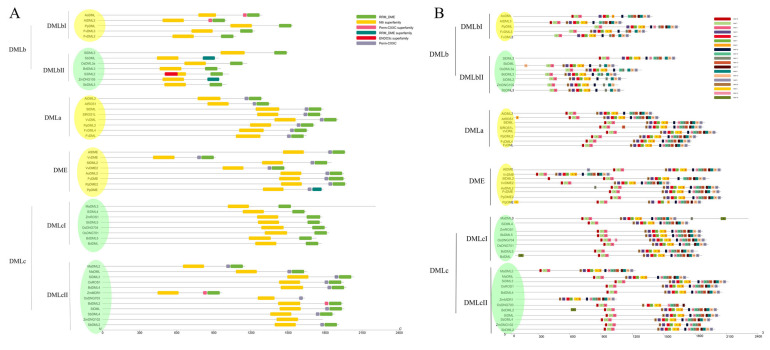
Diagram of major predicted conserved domains and motif patterns in Demeter-like proteins. (**A**) Illustration of conserved domains in Demeter-like proteins. Classes of different genes are displayed on the left. Color green indicates that the class is monocot-specific. Color yellow indicates that the class is eudicot-specific. A scale bar with units of one amino acid from N’ to C’ is shown at the bottom. Domain data were obtained from NCBI CDD. The Locus ID of a gene for its corresponding name can be found in Appendix A. (**B**) Illustration of protein motif patterns of Demeter-like proteins. The rectangular bars represent every single motif, and the number of the motif is placed on the bars. Classes of different genes are displayed on the left and a motif with its number on the right. Color green indicates that the class is monocot-specific. Color yellow indicates that the class is eudicot-specific. A scale bar with units of one amino acid from N’ to C’ is shown at the bottom. Genes in *A. thaliana* are set as control. The Locus ID of a gene for its corresponding name can be found in Appendix A.

**Figure 3 plants-13-02068-f003:**
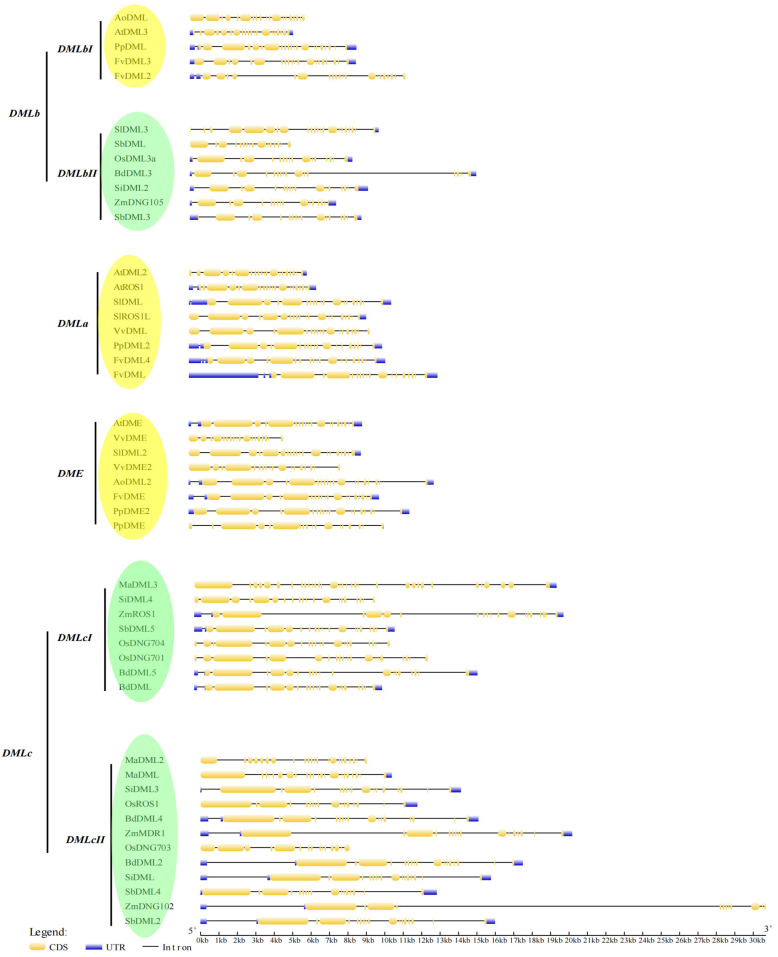
Diagram of gene structure of Demeter-like proteins in 12 land plants. The yellow thick bars stand for exons, and blue rectangular bars stand for UTR, while the thin lines represent introns. Classes of different genes are displayed on the left. Color green indicates that the class is monocot-specific. Color yellow indicates that the class is eudicot-specific. A scale bar with units of bp is shown at the bottom. Genes in *A. thaliana* are set as control.

**Figure 4 plants-13-02068-f004:**
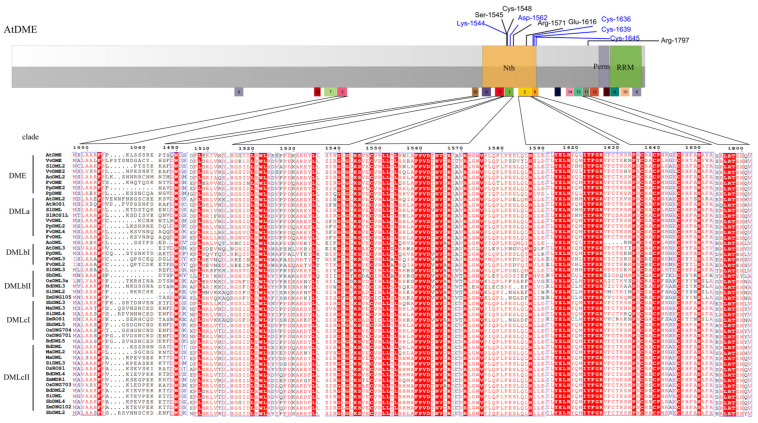
Conserved amino acid sites in 48 Demeter-like proteins. The light-gray- and dark-gray-colored bar at the top is a simplified representative of the AtDME protein, and important conserved sites are designated above the bar. Squares with different color and number below the gray bar represent different motifs of AtDME protein and motif numbers are consistent with those in Figure 2. For the diagram of aligned amino acid sites in the protein sequences, the amino acids designated in white letters and red background are completely conserved among all 48 Demeter-like proteins, and those designated in red letters are partially conserved. Non-conserved sites are designated in black letters.

**Figure 5 plants-13-02068-f005:**
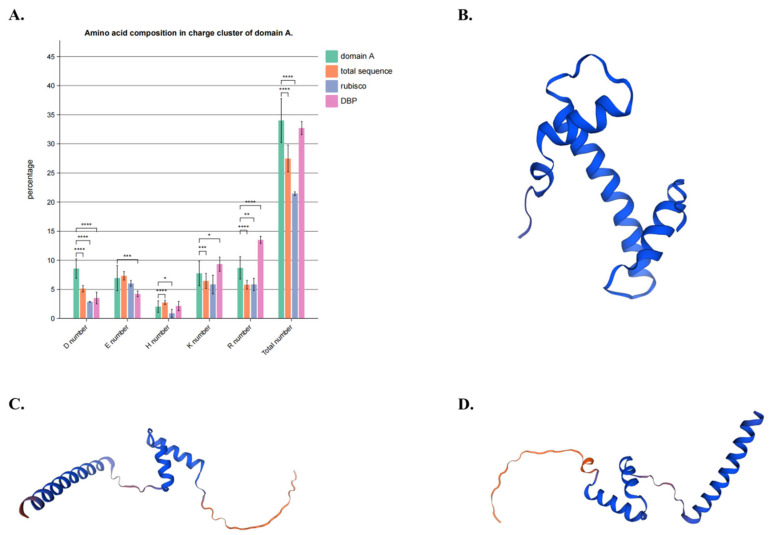
Illustration of potential domain A in Demeter-like proteins. (**A**) Statistics of charge amino acid composition in the potential domain A region. The charge amino acid compositions in entire Demeter-like proteins and rice rubisco are set as negative control, while those of the DNA binding region of the histone-fold domain are set as positive control (designated as DBP). (**B**) Predicted 3-dimensional structure of DBP. (**C**) Predicted 3-dimensional structure of the AtDME potential domain A region. (**D**) Predicted 3-dimensional structure of the SlDML2 potential domain A region. *, **, ***, **** indicate significant differences of charge amino acid composition between different groups: * *p* < 0.05, ** *p* < 0.01, *** *p* < 0.001, **** *p* < 0.0001.

**Figure 6 plants-13-02068-f006:**
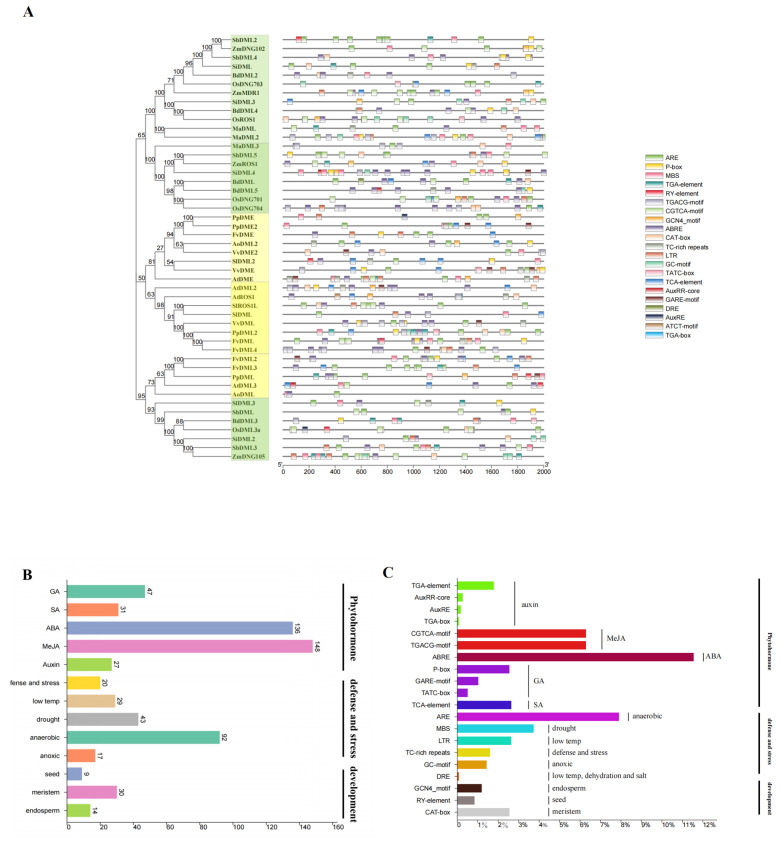
Illustration of phytohormone, defense and stress, and developmental stage response cis-elements in Demeter-like gene promoter region. (**A**) Position of cis-elements in the promoter region of Demeter-like genes in 12 land plants. Colored rectangular bars stand for cis-elements with equal lengths of 40 bp. Classes of different genes are displayed on the left. Color green indicates that the class is monocot-specific. Color yellow indicates that the class is eudicot-specific. A scale bar with units of bp is shown at the bottom. (**B**) Sum-numbers of different cis-elements in response to phytohormone, defense and stress, and developmental stage in Demeter-like gene promoter region of 12 land plants. (**C**) Percentage of total targeted cis-elements in Demeter-like gene promoter region of 12 land plants. GA, gibberellin; MeJA, methyl jasmonic acid; SA, salicylic acid; ABA, abscisic acid; low temp, low temperature.

**Figure 7 plants-13-02068-f007:**
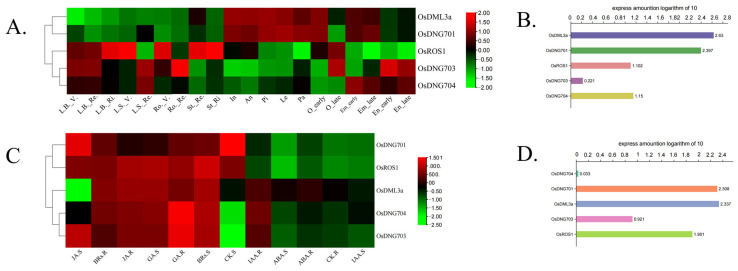
Expression profiles of DNA demethylase genes of different developmental stages and tissues and different hormone treatments. (**A**) Expression heatmap is constructed based on different developmental stages and tissues. The rice DNA demethylase genes are designated on the right. The values in the color scale represent normalized FPKM; red/green indicates the relatively high level/low level of transcript abundance. Tissue types and developmental stages of *Oryza sativa* are indicated at the top of the column. L. leaves; B. blade; S. sheath; Ro. root; St. stem; In. inflorescence; An. anther; Pi. pisti; Le. lemma; Pa. palea; O. ovary; Em. embryo; En. endosperm; Re. reproductive; V. vegetative; Ri. Ripening. (**B**) Relative expression amount of rice DNA demethylases. All data were taken as a logarithm of 10. (**C**) Expression heatmap constructed based on different phytohormone treatments. Maximum alteration was collected as the expression profile. The rice DNA demethylase genes are designated on the right. The values in the color scale represent normalized Cy5/Cy3 ratios. Phytohormone names are indicated at the bottom. JA. represents jasmonic acid; GA. represents gibberellic acid; BR. represents brassinolide; CK. represents cytokinin; ABA. represents abscisic acid; IAA. represents indoleacetic acid; S. represents shoot response; R. represents root response. (**D**) Relative expression amount of rice DNA demethylases. All data were taken as a logarithm of 10.

**Table 1 plants-13-02068-t001:** Identification of Demeter-like proteins in 12 land plants.

	Monocots						Eudicots						Total
	*Musaceae*	*Gramineae*					*Brassicaceae*	*Anacardiaceae*	*Rosaceae*		*Poaceae*	*Vitaceae*	
	Ma	Bd	Os	Sb	Si	Zm	At	Ao	Fv	Pp	Sl	Vv	
Demethylase													
*DMLb* class	0	1	1	2	1	1	1	1	2	1	1	0	12
*DMLa* class	0	0	0	0	0	0	2	0	2	1	2	1	8
*DME* class	0	0	0	0	0	0	1	1	1	2	1	2	8
*DMLc* class	3	4	4	3	3	3	0	0	0	0	0	0	20
Total	3	5	5	5	4	4	4	2	4	4	4	3	48

Bd. represents *Brachypodium distachyon*; Ma represents *Musa acuminata*; Os represents *Oryza sativa*; Sb represents *Sorghum bicolor*; Si represents *Setaria italica*; Zm represents *Zea mays*; Ao represents *Anacardium occidentale*; At represents *Arabidopsis thaliana*; Fv represents *Fragaria vesca*; Pp represents *Prunus persica*; Sl represents *Solanum lycopersicum*; Vv represents *Vitis vinifera*.

**Table 2 plants-13-02068-t002:** K_a_/K_s_ and the divergence time of WGD gene pairs in 12 land plants.

Species	Gene Pair	K_a_/K_s_	Date (Mya)
Ao	AoDML vs. AoDML2	0.192689736	166.914916
Ma	MaDML2 vs. MaDML3	0.232780074	90.99861036
Pp	PpDML vs. PpDML2	0.295172933	108.1189753
Sl	SlROS1L vs. SlDML	0.393649662	34.13422417
Sl	SlROS1L vs. SlDML2	0.188789595	149.3784079
Vv	VvDME vs. VvDML	0.207862231	99.7488283

Ma represents *Musa acuminata*; Ao represents *Anacardium occidentale*; Pp represents *Prunus persica*; Sl represents *Solanum lycopersicum*; Vv represents *Vitis vinifera*. K_a_/K_s_ represents ratio of non-synonymous mutation rate to synonymous mutation rate. Mya represents million years.

**Table 3 plants-13-02068-t003:** Statistics of the light, phytohormone and stress, and defense response cis-elements in Demeter-like gene promoter regions.

Gene Name	Light	Auxin	GA	SA	ABA	MeJA	Defense and Stress	Low Temp	Drought	Seed	Meristem	Endosperm	Anaerobic	Anoxic
*AoDML*	13	0	0	0	1	0	0	0	1	0	0	0	1	0
*AtDML3*	11	1	0	2	3	2	0	0	0	0	0	0	0	0
*PpDML*	13	1	2	0	2	0	0	1	1	1	0	1	2	0
*FvDML3*	5	1	0	2	1	0	0	1	0	0	0	0	6	0
*FvDML2*	16	0	2	2	6	0	1	1	1	0	0	0	4	0
*SlDML3*	9	1	1	0	1	2	1	0	1	0	0	0	1	0
*SbDML*	7	0	1	0	1	4	0	0	0	0	0	0	2	0
*OsDML3a*	7	1	0	0	0	8	0	0	1	1	2	0	3	0
*BdDML3*	10	2	2	0	3	2	0	1	2	0	0	0	0	0
*SiDML2*	15	0	0	0	2	2	0	1	0	0	1	0	1	2
*ZmDNG105*	11	3	0	0	2	8	0	2	2	0	1	0	2	3
*SbDML3*	10	1	0	0	4	4	0	2	2	0	2	0	2	0
*AtDML2*	24	0	0	2	10	2	1	0	0	0	2	1	0	0
*AtROS1*	15	1	0	1	3	4	1	0	0	0	1	1	0	1
*SlDML*	8	1	0	1	2	2	0	1	1	0	0	0	1	0
*SlROS1L*	11	0	0	0	3	4	0	1	0	0	1	0	4	0
*VvDML*	14	0	3	0	4	8	1	0	0	0	0	0	2	0
*PpDML2*	9	3	2	1	3	2	0	0	2	0	2	0	3	1
*FvDML4*	17	0	1	0	8	8	1	1	0	2	1	0	2	0
*FvDML*	13	0	2	0	7	4	1	1	2	3	1	0	3	0
*AtDME*	10	1	1	0	1	4	3	0	2	0	0	1	2	0
*VvDME*	5	0	5	2	1	2	1	1	0	0	1	1	2	0
*SlDML2*	12	0	1	3	1	0	1	0	1	1	0	0	3	0
*VvDME2*	4	0	1	0	1	4	0	0	1	0	2	0	1	0
*AoDML2*	16	0	1	2	2	0	1	0	0	0	0	1	3	0
*FvDME*	11	0	1	1	1	0	0	2	0	0	0	0	2	0
*PpDME2*	14	0	2	2	0	0	1	0	2	0	0	0	2	0
*PpDME*	7	1	0	0	0	0	0	1	1	0	0	1	2	0
*MaDML3*	9	0	0	0	1	6	0	0	1	0	1	0	1	0
*SiDML4*	14	1	4	0	6	2	1	0	5	0	0	1	0	1
*ZmROS1*	14	1	1	1	2	2	0	0	1	0	1	1	0	0
*SbDML5*	9	0	1	1	1	8	0	1	1	0	1	0	4	0
*OsDNG704*	16	1	0	0	6	8	0	1	0	0	2	0	1	0
*OsDNG701*	9	0	0	1	3	6	0	2	1	0	0	1	2	1
*BdDML5*	10	1	1	0	6	0	1	0	0	0	0	0	0	1
*BdDML*	9	1	0	1	4	2	1	1	0	0	1	0	2	0
*MaDML2*	6	0	2	2	1	10	0	1	3	0	1	0	3	1
*MaDML*	7	1	0	0	1	2	1	1	2	0	0	0	1	0
*SiDML3*	8	0	0	1	1	2	0	1	0	0	1	1	2	2
*OsROS1/DNG702*	17	0	1	0	11	6	0	0	0	0	1	1	0	3
*BdDML4*	10	0	1	0	3	2	0	2	0	0	1	0	1	0
*ZmMDR1*	9	0	1	2	3	6	0	1	1	0	0	1	3	0
*OsDNG703*	7	1	2	1	0	0	0	0	0	0	0	0	3	1
*BdDML2*	19	0	0	0	3	4	1	0	1	0	1	0	0	0
*SiDML*	10	1	1	0	0	0	1	1	0	0	1	0	3	0
*SbDML4*	14	0	2	0	11	2	0	0	1	0	1	0	0	0
*ZmDNG102*	7	0	1	0	0	4	0	0	2	0	0	1	3	0
*SbDML2*	7	1	1	0	0	0	0	0	1	1	0	0	7	0

GA represents gibberellin; ABA represents Abscisic Acid; MeJA represents Methyl Jasmonate; SA represents Salicylic acid; Low temp represents low temperature.

**Table 4 plants-13-02068-t004:** Summary of protein properties of Demeter-like proteins in 12 land plants.

	Total No.	Protein Length (aa)	Molecular Weight (kDa)	pI	No. of Genes with pI > 7
*DME* gene family	8	1793 ± 371	196 ± 40	7.5 ± 0.6	7
*DMLa* gene family	8	1680 ± 212	188 ± 22	6.8 ± 0.9	2
*DMLb* gene family	12	1179 ± 210	133 ± 24	7.9 ± 1.0	10
*DMLc* gene family	20	1775 ± 283	197 ± 31	6.5 ± 0.5	1

aa represents amino acid; kDa represents kilo-Dalton; pI represents isoelectric point.

## Data Availability

The datasets used and/or analyzed during the current study are available from the corresponding author on reasonable request.

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
