# Peer review of "Genome-Wide Analysis of DNA Demethylases in Land Plants and Their Expression Pattern in Rice"

_plants, 2024, doi:10.3390/plants13152068_

Round 1

Reviewer 1 Report

Comments and Suggestions for Authors

The manuscript “Genome-Wide Analysis of DNA Demethylases in Land Plants and Their Expression Pattern in Rice” by Mao and co-workers performed the molecular phylogenetic analysis of 48 DNA demethylases in 12 land plant species. Their phylogenetic analysis showed that DNA demethylases in plants can be divided into 4 classes, one of which is specific to dicots and another one specific to monocots. Amino acid alignments and motif searches show potentially important amino acids and motifs conserved among species. These data would be valuable. However, the manuscript includes many major and minor problems as follows.

Major points;

-        The frequencies of cis-elements in Fig.6 and Table 3 does not necessarily means their actual involvement but just only potential involvement. However, the authors claim like the abundance of cis-elements show its actual involvement. It is important to pay attention to draw conclusion. For example, Table 3 shows OsROS1 includes 11 ABA responsive elements (the highest among rice demethylases) but does not respond to ABA at all in Fig 7. Also, their frequency of appearance in all promoters would be different among cis-elements and also among species (GC content can differ among species). Thus, it woluld be informative to present observed/expected data as well.

-        In addition to the above, many other claims are found to be required to pay more attention (i.e. over-discussion). Examples are as follows.

Line 310 The differences…

Line 317 The similar exon-intron structure may come from the lack of experimental verification as they are newly identified. Also the phylogenetic analysis in Fig.1 show that they are less closer.

Line 357 This conserved… I’m not sure the structure in Fig5C,D is similar enough to that of histones to claim this. In addition, even if they bind to DNA, it does not necessarily means their demethylase activity.

Line 412: The transcript level and biological significance does not necessarily correlate.

-        It is not fair to exclude ZmMDR1 in Line 351.

-        The resolution of the figures in Fig.3 is too low. Thus I could not understand the Fig3 part.

-        Line 371: I don’t think it is a good idea to focus on some areas to search for the cis elements because different species are studied in this report. 

-        I am not sure the importance of showing the longest exons in Fig.3. Please explain its importance. If it is, please show it in the scientific ways (i.e. boxplots of the longest exons). 

-        Many incorrect statements. Examples are as follows.

Line 187 four O.sative DNA demethylases : Zemach et al., show 6 demethylases.

Line 217 and others: “WGD” I guess they are discussing about Whole gene duplication. (If they are actually discussing about Whole Genome Duplication and its impacts on the diversification of DNA demethylases, at least please state and compare the dates when Whole Genome Duplication happen, when species differentiation happens, …)

Line 373, 377: our genes

-        Many problems in English writing and also ambiguous expressions. Examples are as follows.

Line 22: What is the functional “sites”?

Line 40: “under certain circumstances”

Line 44: DNA semiconservative replication?  Methylation can be..

Line 71: “genes” to “gene”

Line 192: “Due to …” The claim is ambiguous. Please explain properly.

Suggestions;

I think it is interesting finding that DMLc is monocot-specific and DMLb dicot-specific. In the discussion part, it would be interesting to discuss about it more, especially where they evolutionarily come from, what is their difference and possible biological meaning. Also, it can be discussed whether there is any gene in monocots which are functionally orthologous to DEMETER.

Minor points;

-        The authors put bars to show class information in the figures (Fig1, 2, 3, 4, 6) but it is difficult to read. Please consider to take some other ways to show (i.e. using colors).

-        Many abbreviations are used. Please provide the full words for the first time.

-        Line 183 and other places: NhH-GPD can be a typo of HhH-GPD.

-        Line 48, Line 442 and other places: It sounds like that the authors newly identified 48 DNA demethylase genes, but actually many of them were already identified and characterized. Please specify correctly.

-        Inappropriate citations are found.

-        Please put the unit for the Date in Table 2. I guess it is “My”.

-        The authors use different words to express domains (i.e. DME domain, NhH-GPD domain, domain A, glycosylase domain, RRM_DME domain, Nth domain, Perm-CXXC, permease domain, some of which may overlap), which may confuse the readers. At least, please explain their relationship.  

-        For the readers’ convenience, Please put their gene names as well in Fig 1 as the authors use gene names only in other Figs.

-        Please briefly explain the domains present in Fig2. What is the difference between RRM-DME and RRM-DME superfamily?

-        It is difficult to understand the abbreviations in Fig 7A.

Comments on the Quality of English Language

Moderate editing of English language is required.

Author Response

Response to reviewer 1

The manuscript “Genome-Wide Analysis of DNA Demethylases in Land Plants and Their Expression Pattern in Rice” by Mao and co-workers performed the molecular phylogenetic analysis of 48 DNA demethylases in 12 land plant species. Their phylogenetic analysis showed that DNA demethylases in plants can be divided into 4 classes, one of which is specific to dicots and another one specific to monocots. Amino acid alignments and motif searches show potentially important amino acids and motifs conserved among species. These data would be valuable. However, the manuscript includes many major and minor problems as follows.

  1. The frequencies of cis-elements in Fig.6 and Table 3 does not necessarily means their actual involvement but just only potential involvement. However, the authors claim like the abundance of cis-elements show its actual involvement. It is important to pay attention to draw conclusion. For example, Table 3 shows OsROS1 includes 11 ABA responsive elements (the highest among rice demethylases) but does not respond to ABA at all in Fig 7. Also, their frequency of appearance in all promoters would be different among cis-elements and also among species (GC content can differ among species). Thus, it would be informative to present observed/expected data as well.

Response: Thank you for your insightful comments. We acknowledge the importance of distinguishing between potential and actual involvement of cis-elements as indicated in the analysis. We have taken your feedback into account and have made the following revisions to our manuscript: We have incorporated additional statements to clarify that the presence and frequency of cis-elements in Fig. 6 and Table 3 are suggestive of potential involvement rather than definitive proof of their role. This distinction is now emphasized to ensure that our conclusions are not misinterpreted. We have also revised the text to highlight the discrepancy between the predicted involvement of cis-elements and the experimental outcomes observed in rice, as exemplified by the OsROS1 case. This is now clearly stated in the manuscript to emphasize the complexity of gene regulation and the limitations of predictive analyses. Specifically, we have modified the text on page 21, section 4.3, lines 560-573, to include phrases such as "latent and possible" to denote the speculative nature of the involvement of cis-elements. Additionally, the example provided by the reviewer regarding OsROS1 has been integrated into the manuscript on page 21, section 4.4, lines 603-612. This serves to illustrate that a high abundance of cis-elements does not necessarily correlate with a response to ABA, reinforcing the concept of potential rather than actual involvement. We believe these revisions enhance the manuscript by providing a clearer explanation and a more cautious interpretation of the data presented.

  1. In addition to the above, many other claims are found to be required to pay more attention (i.e. over-discussion). Examples are as follows.
    • Line 310 The differences…

Response: Thank you for bringing this to our attention. After review, we have determined that the sentence in question may be too detailed for section 3.4. Therefore, we have decided to remove it from the article.

The similar exon-intron structure may come from the lack of experimental verification as they are newly identified. Also the phylogenetic analysis in Fig.1 show that they are less closer.

Response: Thank you for pointing this out. We agree with this comment. So we have added a statement in page 19, section 4.1, that the expected gene structure generated from bioinformatical algorithm may need further modification through experiments.

  • Line 357 This conserved… I’m not sure the structure in Fig5C, D is similar enough to that of histones to claim this. In addition, even if they bind to DNA, it does not necessarily mean their demethylase activity.

Response: Thank you for pointing this out. We agree with this comment. We think this structure may be crucial for DNA-protein interaction and DNA binding instead of DNA demethylase activity, so we have changed the origin expression into “...may be necessary for protein-DNA interaction between DNA and DNA demethylases.” in page 12, section 3.5. But from the figure, the structure does feature a traditional helix-hairpin-helix pattern, which binds DNA.

  • Line 412: The transcript level and biological significance does not necessarily correlate.

Response: Thank you for pointing this out. We agree with this comment. So we have already removed this statement.

  • It is not fair to exclude ZmMDR1 in Line 351.

Response: Thank you for pointing this out. We agree with this comment. Therefore, we have added it into the survey and remade our graph, and also deleted the statement in the article.

  • Line 371: I don’t think it is a good idea to focus on some areas to search for the cis elements because different species are studied in this report.

Response: Thank you for pointing this out. We agree with this comment. So we only propose directions on areas that seem to have more potential, which means due to our statistics there is a relative high frequency of correspondent cis-element across all 12 species or most of them (Table 3; example as page 14, section 3.6). And in another article, the same statistical method was performed (Hou et al., 2019), showing the potential of cis-elements survey.

  • I am not sure the importance of showing the longest exons in Fig.3. Please explain its importance. If it is, please show it in the scientific ways (i.e. boxplots of the longest exons).

Response: Thank you for pointing this out. But actually, we are not only showing the longest exons. We are showing the intron-exon pattern. The core point is the pattern of intron-exon instead of the length, and the authors mentioned the length only to make it more clear to see the intron-exon pattern among different genes.

  1. Many incorrect statements. Examples are as follows

  • Line 187 four O.sative DNA demethylases : Zemach et al., show 6 demethylases.

Response: Thank you for pointing this out. We partially agree with this statement. But in our identification and classification, only four of them feature two complete domains (Nth domain and DME domain), which are required to become DNA demethylases in our study. The other two do not contain DME domain and a complete Nth domain, thus, we do not consider them as DNA demethylases like DNA demethylases in A. thaliana in our study.

  • Line 217 and others: “WGD” I guess they are discussing about Whole gene duplication. (If they are actually discussing about Whole Genome Duplication and its impacts on the diversification of DNA demethylases, at least please state and compare the dates when Whole Genome Duplication happen, when species differentiation happens, …)

Response: Thank you for pointing this out. In the article, it was mentioned that WGD is Whole Genome Duplication. And we have compared the date (70 millions years ago) of differentiation of monocots and dicots and the date of whole genome duplication events(page 8, section 3.2).  

  • Line 373, 377: our genes

Response: Thank you for pointing this out. We agree with this statement. We have changed “our genes” to “48 genes”.

  1. Many problems in English writing and also ambiguous expressions. Examples are as follows.

  • Line 22: What is the functional “sites”?

Response: Thank you for pointing this out. We agree with this statement. In the line, sites mean the catalytic amino acids. And we have found this word inappropriate, and changed it to “regions” which contains the meaning of both functional domains and catalytic amino acids.

4.2 Line 40: “under certain circumstances”

Response: Thank you for pointing this out. We agree with this statement. We have changed it to “for correspondent DNA contexts”.

4.3 Line 44: DNA semiconservative replication?  Methylation can be..

Response: Thank you for pointing this out. We agree with this statement. And we have further explained the mechanism: “DNA semiconservative replication allows passive DNA demethylation to automatically occur in newly synthesized strands, without the help of addition enzymes or factors. In newly synthesized DNA strands, all the bases lack any kinds of modification like methylation. Thus, if there exists no specific enzymes to create methylation on the newly synthesized strands and maintain DNA methylation level in cell, DNA methylation will soon fade away after several rounds of DNA replication.”

4.4 Line 71: “genes” to “gene”

Response: Thank you for pointing this out. We have completed the alternation.

4.5 Line 192: “Due to …” The claim is ambiguous. Please explain properly.

Response: Thank you for pointing this out. And we have add an further explanation “...and far relationships between some genes.” into the sentence.

  1. Suggestions;

I think it is interesting finding that DMLc is monocot-specific and DMLb dicot-specific. In the discussion part, it would be interesting to discuss about it more, especially where they evolutionarily come from, what is their difference and possible biological meaning. Also, it can be discussed whether there is any gene in monocots which are functionally orthologous to DEMETER

Response: Thank for pointing this out. We agree with this comment. Therefore, we have further investigated into the evolutionary details of those species and have made some assumptions about the evolutionary origin of some DNA demethylases. Detailed discussions can be found in section 4.1.

  1. Minor points
    • The authors put bars to show class information in the figures (Fig1, 2, 3, 4, 6) but it is difficult to read. Please consider to take some other ways to show (i.e. using colors).

Response: Thank you for pointing this out. We agree with this comment. And improved figures were made, and colors were added.

  • Many abbreviations are used. Please provide the full words for the first time.

Response: Thank you for pointing this out. And we have provided those extensions for them in the article.

  • Line 183 and other places: NhH-GPD can be a typo of HhH-GPD.

Response: Thank you for pointing this out. We agree with this comment. And we have changed it. 

  • Line 48, Line 442 and other places: It sounds like that the authors newly identified 48 DNA demethylase genes, but actually many of them were already identified and characterized. Please specify correctly.

Response: Thank you for pointing this out. We agree with this comment. And we have made some improved statements.

  • Inappropriate citations are found.

Response: Thank you for pointing this out. To solve this, we have carefully checked our references and citations again, and fixed all the issues.

  • Please put the unit for the Date in Table 2. I guess it is “My”.

Response: Thank you for pointing this out. We agree with this statement. And we have already added the unit. We have changed it to “Mya” which means million years ago. 

6.7 The authors use different words to express domains (i.e. DME domain, NhH-GPD domain, domain A, glycosylase domain, RRM_DME domain, Nth domain, Perm-CXXC, permease domain, some of which may overlap), which may confuse the readers. At least, please explain their relationship.  

Response: Thank you for pointing this out. We agree with this statement. We have added some short explanations, for example, page 8, section 3.2 “...all demethylases possess the Nth domain (COG0177), and almost all have the RRM_DME Domain (pfam15628). Nth domain is an endonuclease III domain, which is generally predicted to be responsible for DNA replication, recombination, and repair; while RRM-DME domain is believed to be specified for DML proteins and is predicted to facilitate the recognition to specific DNA sequence through the guide of ssDNA or RNA (Iyer et al., 2011; Wang et al., 2023).”.

6.8 For the readers’ convenience, Please put their gene names as well in Fig 1 as the authors use gene names only in other Figs.

Response: Thank you for pointing this out. We agree with this comment. We have adjusted the figure.

6.9 Please briefly explain the domains present in Fig2. What is the difference between RRM-DME and RRM-DME superfamily?

Response: Thank you for pointing this out. The difference is superfamily is more general and has less conservativeness in sequence.

6.10  It is difficult to understand the abbreviations in Fig 7A.

Response: Thank you for pointing this out. We agree with this comment. So we have already added extensions for the abbreviations in the legend of Fig 7A.

Comments on the Quality of English Language

Moderate editing of English language is required.

Response: Thank you for pointing this out. We have carefully checked our language and performed English editing.

Reviewer 2 Report

Comments and Suggestions for Authors

The research addresses an important topic and currently aligns with the preferred trend of investigating the impact of epigenetic factors' variability on the development of better crop varieties in terms of agronomic traits. The article is suitable for publication, but it needs to be revised according to the following guidelines.

Introduction

Lines 39-42 – instead of  “For the maintenance of different methylated sequence context, plants utilize different enzymes under certain circumstances (for example, MET1 for CG and CMT3 for CHG) (Shim et al., 2021), and the methylation at the asymmertic site CHH must be established de novo (Zhu, 2009).”, should be “For the maintenance of different methylated sequence context, plants utilize different enzymes under certain circumstances (for example, MET1 for CG and CMT3 for CHG) (Shim et al., 2021), and the methylation at the asymmertic site CHH (for example, CMT2 with DRM2) must be established de novo (Zhu, 2009).”

No shortcut extensions, for example: MET1, CMT3, DRM, ROS1, DME, DML2, DML3;

Lines 43-45 – ”DNA semiconservative replication allows passive DNA demethylation to automatically occur in newly synthesized strands.” – how, please add 2-3 sentences;

55-56 – “After that, other demethylases in A. thaliana were identified (Choi et al., 2002).” This sentence is redundant and can be removed, or it should be supplemented with information about other DNA demethylases."

Lines 103-104 – „However, resent studies on specific DNA demethylases mainly focus on A. thaliana and O. sativa.- authors of the studies should be cited in parentheses.

Materials and Methods

Line 157 – Hou et al., 2021 should be included in the references;

Results
Lines 203-206 – Latin names should be italicized throughout the document;

Lines 214-217, 222-224, 320-328, 369-373, 414-415 – These are results discussion.

Lines 236-238 – All demethylases possess the Nth domain, and almost all have the RRM-DME domain.

Lines  247-249 - The sentence, Two enzymes (AtDML3 and one in A.occidentale) from eudicot in the DMLb group are expected to have 248 the Perm-CXXC domain, while others in that class are not.”, will be more understandable: “Two enzymes (AtDML3 and AoDML) from eudicot in the DMLb group are expected to have 248 the Perm-CXXC domain, while others in that class are not.”

Line 252 – after (Figure.2)., add: In DMLcII, all possess three domains, except OSDNG703 and ZmDNGI02.

The legend for Figure 2 is unclear and needs revision; the Perm-CXXC domain line for SIDML3, which should be placed below.

Figure 3, both the diagram and the legend, are also unclear and need to be made more legible.

 The legend for Figure 2 is unclear and needs revision, especially the Perm-CXXC domain line for SIDML3, which should be placed below.

 Dyskusja

Lines 469-472 – instead of   A high frequency of the appearance of charge amino acids in this region also indicates that this region is very likely to play a role in DNA binding (Mok et al., 2010; Figure.5A), which may play an important role in the stabilization or recognition of DNA substrates.”, should be “A high  frequency of the appearance of charge amino acids in this region also indicates that this 470 region is very likely to play a role in DNA binding (Figure.5A) (Mok et al., 2010), which may play an important role in the stabilization or recognition of DNA substrates.”

Subsection 4.4 should be supplemented with information on how the expression of DNA demethylases changes in other species, such as ROS1 in A. thaliana, in response to specific hormones and in different tissues. Additionally, information about other DNA demethylases should be included to provide a more comprehensive overview.

The discussion should conclude with a summary. The main findings should be highlighted: the conducted studies on cis-promoter elements indicated possible signaling and regulatory pathways for DNA demethylases. Furthermore, the expression profile of DNA demethylases across different developmental stages, tissues, and in response to stress and various phytohormonal signals was shown. There is a lack of examples demonstrating the crucial role of DNA demethylases in creating better crop varieties in terms of agronomic traits

References

Please check the literature references; many references are missing, e.g. Shim et al. 2021; Tsukado et al. 2006; Kapoor et al. 2005; Huh et al. 2008; Zhu and Kapoor et al. 2007, should by Zhu et al. 2007;

whether it should be Park et al. 2016 or 2017, Du et al. 2022 or 2023?

Author Response

Response to reviewer 2

The research addresses an important topic and currently aligns with the preferred trend of investigating the impact of epigenetic factors' variability on the development of better crop varieties in terms of agronomic traits. The article is suitable for publication, but it needs to be revised according to the following guidelines.

1.Introduction

1.1Lines 39-42 – instead of  “For the maintenance of different methylated sequence context, plants utilize different enzymes under certain circumstances (for example, MET1 for CG and CMT3 for CHG) (Shim et al., 2021), and the methylation at the asymmertic site CHH must be established de novo (Zhu, 2009).”, should be “For the maintenance of different methylated sequence context, plants utilize different enzymes under certain circumstances (for example, MET1 for CG and CMT3 for CHG) (Shim et al., 2021), and the methylation at the asymmertic site CHH (for example, CMT2 with DRM2) must be established de novo (Zhu, 2009).”

Response: Thank you for pointing this out. We agree with this comment. So we have changed the original sentence at page 1, section 1, line 40-44.

1.2No shortcut extensions, for example: MET1, CMT3, DRM, ROS1, DME, DML2, DML3;

Response: Thank you for pointing this out. We agree with this comment. We have given correspondent extensions for this genes. For example, in page 2, section 1, line 55-56 (Demeter for DME...).

1.3 Lines 43-45 – ”DNA semiconservative replication allows passive DNA demethylation to automatically occur in newly synthesized strands.” – how, please add 2-3 sentences;

Response: Thank you for pointing this out. We agree with this comment. We have provided detailed mechanism of passive DNA demethylation in page 2, section 1, line 47-51: DNA semiconservative replication allows passive DNA demethylation to automatically occur in newly synthesized strands, without the help of addition enzymes or factors. In newly synthesized DNA strands, all the bases lack any kinds of modification like methylation. Thus, if there exists no specific enzymes to create methylation on the newly synthesized strands and maintain DNA methylation level in cell, DNA methylation will soon fade away after several rounds of DNA replication.

1.4 55-56 – “After that, other demethylases in A. thaliana were identified (Choi et al., 2002).” This sentence is redundant and can be removed, or it should be supplemented with information about other DNA demethylases."

Response: Thank you for pointing this out. We agree with this comment. And we have removed this sentence in the article.

1.5 Lines 103-104 – „However, resent studies on specific DNA demethylases mainly focus on A. thaliana and O. sativa.” - authors of the studies should be cited in parentheses.

Response: Thank you for pointing this out. We agree with this comment. We have added necessary citations after that statement in page 3, section 1, line 110.

  1. Materials and Methods

Line 157 – Hou et al., 2021 should be included in the references;

Response: Thank you for pointing this out. We agree with this comment. We have added the citation in the references.

3.Results

3.1Lines 203-206 – Latin names should be italicized throughout the document;

Response: Thank you for pointing this out. We agree with this comment. We have carefully checked our Latin names and corrected them.

3.2 Lines 214-217, 222-224, 320-328, 369-373, 414-415 – These are results discussion.

Response: Thank you for pointing this out. We agree with this comment. So some inappropriate sentences have been removed and some have been replaced to the discussion part. For example, we placed the sentences in section 3.4 that “For example, the class DMLc has...is relatively conserved.” in section 4.1 and made modifications.

3.3 Lines 236-238 – All demethylases possess the Nth domain, and almost all have the RRM-DME domain.

Response: Thank you for pointing this out. We have made this correlation.

3.4 Lines  247-249 - The sentence, ”Two enzymes (AtDML3 and one in A.occidentale) from eudicot in the DMLb group are expected to have 248 the Perm-CXXC domain, while others in that class are not.”, will be more understandable: “Two enzymes (AtDML3 and AoDML) from eudicot in the DMLb group are expected to have 248 the Perm-CXXC domain, while others in that class are not.”

Response: Thank you for pointing this out. We agree with this comment. We have changed this sentence as you suggested, and made it more understandable in page 8, section 3.3.

3.5 Line 252 – after (Figure.2)., add: In DMLcII, all possess three domains, except OSDNG703 and ZmDNGI02.

Response: Thank you for pointing this out. We agree with this comment. We have added this sentence in page 8, section 3.3.

3.6 The legend for Figure 2 is unclear and needs revision; the Perm-CXXC domain line for SIDML3, which should be placed below.

Response: Thank you for pointing this out. We agree with this comment. We have made some modifications on our figures and their legends, trying to make them more clear and corrected some small mistakes like this in Figure 2.

3.7 Figure 3, both the diagram and the legend, are also unclear and need to be made more legible.

Response: Thank you for pointing this out. We agree with this comment. We have made some modifications on our figures and their legends, trying to make them more clear.

4.Discussion

4.1Lines 469-472 – instead of   “A high frequency of the appearance of charge amino acids in this region also indicates that this region is very likely to play a role in DNA binding (Mok et al., 2010; Figure.5A), which may play an important role in the stabilization or recognition of DNA substrates.”, should be “A high  frequency of the appearance of charge amino acids in this region also indicates that this 470 region is very likely to play a role in DNA binding (Figure.5A) (Mok et al., 2010), which may play an important role in the stabilization or recognition of DNA substrates.”

Response: Thank you for pointing this out. We agree with this comment. And we have changed the place and structure of the brackets in the sentence.

4.2Subsection 4.4 should be supplemented with information on how the expression of DNA demethylases changes in other species, such as ROS1 in A. thaliana, in response to specific hormones and in different tissues. Additionally, information about other DNA demethylases should be included to provide a more comprehensive overview.

Response: Thank you for pointing this out. We agree with this comment. So we have searched for more reports in this field and added the results from some of them (Yu et al., 2013; Zhang et al., 2022) in our discussion (page 21, section 4.4, line 607-612).

4.3 The discussion should conclude with a summary. The main findings should be highlighted: the conducted studies on cis-promoter elements indicated possible signaling and regulatory pathways for DNA demethylases. Furthermore, the expression profile of DNA demethylases across different developmental stages, tissues, and in response to stress and various phytohormonal signals was shown. There is a lack of examples demonstrating the crucial role of DNA demethylases in creating better crop varieties in terms of agronomic traits.

Response: Thank you for pointing this out. We agree with this comment. We have added a new section, section 5 conclusion, in our article. And the main findings are highlighted: “Cis-element analysis of promoter region in DNA demethylase genes provided future directions in studying the role of DNA demethylases in stress and defense response or in response to phytohormone, while expression profile construction in rice, to some extent, support the general role of DNA demethylases in organisms through pointing out their expression alternation and difference in response to different phytohormones and in different tissues.”

5.References

Please check the literature references; many references are missing, e.g. Shim et al. 2021; Tsukada et al. 2006; Kapoor et al. 2005; Huh et al. 2008; Zhu and Kapoor et al. 2007, should by Zhu et al. 2007;

whether it should be Park et al. 2016 or 2017, Du et al. 2022 or 2023?

Response: Thank you for pointing this out. We agree with this comment. So we have carefully checked our references and citations again, and added the missing references and corrected the incorrect citations.

Reviewer 3 Report

Comments and Suggestions for Authors

The manuscript by Mao, Xiao et al. titled "Genome-wide analysis of DNA demethylases in land plants and their expression pattern in rice" explores the phylogentic and structual relationship between DNA demethylases (glycosylases) across 12 land plants, most of which are important crops, from both monocot and eudicot clades. Through these appraches the Authors identify novel evolutionary relationships between 48 DNA demethylases, including some classified de novo in this study, which yield 4 main classes, some of which specific to either monocot or eudicot clade. In recent years, many studies in the model plant Arabidopsis have highlighted the key role of  DNA glycosylases and active DNA methylation in phenotypic plasticity and responce to fluctutating environmental conditions, such as abiotic and biotic stresses. This work will be of interest for future translational research aming to  study these processes in the crop species investigated in this study.  

The paper is well understandable but not clearly written, and could benefit from English editing throughout its text. The methodologies are clearly explained. The manuscript would also need some changes to figures and text, as detailed below: 

Major corrections: 

- Introduction, lines 93-99: For the model plant Arabidopsis there are many studies assessing the role of active DNA demethylation in plant-microbe interaction. This is particularly relevant to observation in this study about expression of rice DNA demethylases under different phytohormone treatments (Figure 7C and D), which are involved in both abiotic but also biotic stress resistance. The Authors should add a paragraph in the introduction discussing the role of DNA demethylases (particularly ROS1) in pathogen plant-microbe interaction in Arabidopsis, in addition to the cited example in N. benthamiana, given its importance in this context.

- Figure 2 and 3: The order of Figure panels for Figure 2 and 3 does not fit well with the text of the manuscript. Given how Results sections 3.3 and 3.4 are listed, Figure 3B should become panel B of Figure 2 (current Figure 2 would become panel A of the new 2-panel Figure 2). Figure 3 should only contain the current panel A, which should also be increased to at least  twice the size to improve readability of gene names and visibility of exon/intron structures. 

- Figure 7: Results from panels C and D are barely referenced in the "Results" or "Discussion" section of the main text. This is a missed opportunity because there seems to be a very clear pattern of downregulation for all rice DNA demethylases under auxin, ABA or citokinin treatments, and upregulation in the rest. This is a quite interesting result. Additionally, the Authors mention the effect of phytohormones in the abstract, whithout any mention in the Results or Discussion sections. New paragraphs describing these results in more details in Results section 3.7 and Discussion section 4.4 are needed. 

Minor corrections (optional comments are not critical but would, in my opinion, improve the manuscript slightly): 

Figures:

- Figure 1 (optional): a small icon next to each gene representing if a gene belongs to the monocot or eudicot clade would really help visualize the fact that DMLc class and DMLbII subclass are monocot specific. 

- Figure 2: increase legend size. as mentioned above, Figure 3B should be merged with Figure 2 to become Figure 2B. 

- Figure 3: as mentioned above, move panel 3B to Figure 2, and increase size of Figure 2A to at least twice the current size to improve readability. For current panel 3B, increase legend size to improve readability.

- Figure 5: Panels C and D are never mentioned in the text. Please add a sentence about them. 

-Figure 6 (optional): In Figure 6A the order of genes could be the same of previous Figures (like Figure 1, 2 and 3), based on their Phylogenetic relationship.

Text:

- Abstract, line 21: mention that the expression profiles were assesses for rice DNA demethylases specifically.

- Introduction, lines 30-53: While overall the manuscript could benefit from English editing, this section is really hard to read and to understand. Please edit significantly. 

- Introduction, line 51: please define acronym AP in the text (apurinic/apyrimidinic site).

- Introduction: please spell out the acronyms DME DNG and DML the first time they are mentioned.

- Discussion, line 474-475: add reference to Figure 5B (and C and D?)

Supplementary:

- Table.S1: some lines are not correctly aligned (ie gene names from one species are listed under the next species in the line below). Please revise.

Comments on the Quality of English Language

The Manuscript is well understandable but not clearly written, and could benefit from English editing throughout its text. 

Particularly, the intial paragraph of the Introduction is really hard to read. Also,  there are minor grammatical mistakes across the main text. 

Author Response

Response to reviewer 3

The paper is well understandable but not clearly written, and could benefit from English editing throughout its text. The methodologies are clearly explained. The manuscript would also need some changes to figures and text, as detailed below:

Major corrections:

  1. Introduction, lines 93-99: For the model plant Arabidopsis there are many studies assessing the role of active DNA demethylation in plant-microbe interaction. This is particularly relevant to observation in this study about expression of rice DNA demethylases under different phytohormone treatments (Figure 7C and D), which are involved in both abiotic but also biotic stress resistance. The Authors should add a paragraph in the introduction discussing the role of DNA demethylases (particularly ROS1) in pathogen plant-microbe interaction in Arabidopsis, in addition to the cited example in N. benthamiana, given its importance in this context.

Response: Thank you for pointing this out. We agree with this comment. So we have searched for more information on this field and found more examples. Consequently, we have added an additional paragraph in introduction discussing pathogen plant-microbe interactions (page 3, section 1, line 111-133): “Besides all those biological roles of DNA demethylase, it is also important to notice that ... But whether the expression level of DNA demethylases alters in response to specific phytohormone remains unclear.” 

  1. Figure 2 and 3: The order of Figure panels for Figure 2 and 3 does not fit well with the text of the manuscript. Given how Results sections 3.3 and 3.4 are listed, Figure 3B should become panel B of Figure 2 (current Figure 2 would become panel A of the new 2-panel Figure 2). Figure 3 should only contain the current panel A, which should also be increased to at least twice the size to improve readability of gene names and visibility of exon/intron structures.

Response: Thank you for pointing this out. We agree with this comment. So we have changed the panels and legends of Figure 2 and 3.

  1. Figure 7: Results from panels C and D are barely referenced in the "Results" or "Discussion" section of the main text. This is a missed opportunity because there seems to be a very clear pattern of downregulation for all rice DNA demethylases under auxin, ABA or citokinin treatments, and upregulation in the rest. This is a quite interesting result. Additionally, the Authors mention the effect of phytohormones in the abstract, whithout any mention in the Results or Discussion sections. New paragraphs describing these results in more details in Results section 3.7 and Discussion section 4.4 are needed.

Response: Thank you for pointing this out. We agree with this comment. So we have further analyzed our results and discussed more about the role of phytohormone. New paragraph in section 3.7 (From the results, it is clearly illustrated that..in rice roots under the treatment of IAA (Figure.7C).) and section 4.4 (The next heatmap (Figure 7C) illustrates that...further studied through detailed mechanism investigations) are added.

  1. Minor corrections (optional comments are not critical but would, in my opinion, improve the manuscript slightly):

Figures:

  • Figure 1 (optional): a small icon next to each gene representing if a gene belongs to the monocot or eudicot clade would really help visualize the fact that DMLc class and DMLbII subclass are monocot specific.

Response: Thank you for pointing this out. We have added different colors to represent monocot-specific or dicot-specific.

Figure 2: increase legend size. as mentioned above, Figure 3B should be merged with Figure 2 to become Figure 2B.

Response: Thank you for pointing this out. We agree with this comment. The figures were changed and improved.

4.2 Figure 3: as mentioned above, move panel 3B to Figure 2, and increase size of Figure 2A to at least twice the current size to improve readability. For current panel 3B, increase legend size to improve readability.

Response: Thank you for pointing this out. We agree with this comment. The figures were changed and improved.

4.3 Figure 5: Panels C and D are never mentioned in the text. Please add a sentence about them.

Response: Thank you for pointing this out. We agree with this comment. And we have added a sentence about them in page 12, section 3.5.

4.4 Figure 6 (optional): In Figure 6A the order of genes could be the same of previous Figures (like Figure 1, 2 and 3), based on their Phylogenetic relationship.

Response: Thank you for pointing this out. But actually, the phylogenetic tree remains the same as the other figures, and only the order of how genes are displayed was changed in the Figure.

Text:

4.5 Abstract, line 21: mention that the expression profiles were assesses for rice DNA demethylases specifically.

Response: Thank you for pointing this out. We agree with this comment. So we have added “rice” in page 1.

4.6  Introduction, lines 30-53: While overall the manuscript could benefit from English editing, this section is really hard to read and to understand. Please edit significantly.

Response: Thank you for pointing this out. We agree with this comment. English editing has been performed.

4.7 Introduction, line 51: please define acronym AP in the text (apurinic/apyrimidinic site).

Response: Thank you for pointing this out. We agree with this comment. The definition has been added in introduction.

4.8 Introduction: please spell out the acronyms DME DNG and DML the first time they are mentioned.

Response: Thank you for pointing this out. We agree with this comment. We have given correspondent extensions for this genes.

4.9  Discussion, line 474-475: add reference to Figure 5B (and C and D?)

Response: Thank you for pointing this out. We agree with this comment. The references have been added in section 4.2.

Supplementary:

4.10 Table.S1: some lines are not correctly aligned (ie gene names from one species are listed under the next species in the line below). Please revise.

Response: Thank you for pointing this out. We agree with this comment. And we have revised the table.

comments on the Quality of English Language

The Manuscript is well understandable but not clearly written, and could benefit from English editing throughout its text.

Particularly, the intial paragraph of the Introduction is really hard to read. Also,  there are minor grammatical mistakes across the main text.

Response: Thank you for pointing this out. We have performed some English editing for the article and made it more understandable.

Round 2

Reviewer 1 Report

Comments and Suggestions for Authors

Thank you for revising the manuscript. The manuscript has been extensively improved.

Reviewer 3 Report

Comments and Suggestions for Authors

The Authors answered all my comments.

Comments on the Quality of English Language

The Authors have improved English language in some parts, but the manuscript would still benefit from additional edits throughout to improve readability